# Analysis on the Influence of Industrial Structure on Energy Efficiency in China: Based on the Spatial Econometric Model

**DOI:** 10.3390/ijerph20032134

**Published:** 2023-01-24

**Authors:** Xin Zheng, Zi Ye, Zhong Fang

**Affiliations:** 1Department of Cultural Industry, Concord University College, Fujian Normal University, Fuzhou 350100, China; 2School of Economics, Fujian Normal University, Fuzhou 350007, China

**Keywords:** industrial structure, energy efficiency, spatial Durbin model

## Abstract

Compared with other developed countries, China’s energy efficiency level is not optimal, but it has indeed made remarkable achievements in its long-term development, mainly due to efforts targeting the adjustment of industrial structure. This research, therefore, uses a spatial econometric model to study the energy efficiency of 30 provinces in China with data from the panel from 2004 to 2019, and studies the impact of industrial structure on energy efficiency from the overall sample, for different time periods and across the three regional scales of eastern, central and western regions. The following conclusions are drawn from the empirical analysis. (1) China’s energy efficiency indicators have significant geographic spatial correlation and regional spatial structure differences. (2) In the full sample condition, the industrial structure has a positive impact on the energy efficiency of China’s provinces, but it also shows a significant negative spatial spillover effect. (3) Industrial structure was positively correlated with energy efficiency from 2004 to 2011. (4) The industrial structure in the east promotes energy efficiency, while the industrial structure in the central and western regions inhibits energy efficiency improvement. (5) Government intervention and scientific and technological innovation have had a spatial impact on energy efficiency in China’s provinces, while marketization and the average income of residents have had no significant impact.

## 1. Introduction

Energy utilization is an inevitable part of social production. Energy efficiency is not only an important indicator that reflects an economy’s sustainable development potential, but is also an important factor to improve the nation’s overall economic competitiveness. Over the past 40 years of reform and opening up, China’s overall economy has grown by more than 20 times to now become the second largest in the world. However, as China has been in the development stage of high investment and high energy consumption for a long time, it also is the largest energy demander and air pollution emitter globally [1]. According to the Statistical Review of World Energy 2021, the international energy structure is facing a huge impact, driven by COVID-19 and international tensions, and under the background of emphasis on secondary energy the characteristics of coal energy consumption in China have gradually changed. China is in an economic transition stage, with increasingly strong energy demand and huge pressure on energy supply. According to the latest China Energy Economic Index, China’s energy economy will recover steadily in 2022, showing a similar cycle to the macro economy, but the fluctuation range is small, acting as a buffer zone. In 2023, the trend of China’s energy economy will be better, and new energy and energy integration industries will usher in new development. Because the characteristics of its coal energy consumption are difficult to change in a short period of time, it is particularly critical to improve the mode of energy consumption in the production phase, promote utilization efficiency and control the excessive growth of energy consumption. In recent years, China has been committed to enhancing the process of industrial structure optimization, energy resource distribution system marketization, and energy usage. Despite all this, from the perspective of international comparison, China’s energy efficiency is still low, and its proportion of renewable energy is too small. There are also large differences between regions [2], and so the sustainable development of energy has a long way to go.

There are three kinds of research on energy efficiency in this paper. First, for the calculation of energy efficiency, data envelopment analysis (DEA) has been widely used since it was proposed. Shen [3] put the energy efficiency values of 30 provinces in China, measured by super-efficient DEA, into the Moran index analysis framework and found that provincial energy efficiency in China exhibits spatial autocorrelation. The process of using DEA can complement the research objectives with the help of other models. For example, Wang [4] used DEA–Malmquist index to measure total factor energy efficiency and decomposed it into three parts for empirical analysis. In addition to DEA model, Yu et al. [5] took the Malmquist–Luenberger index method to measure total factor energy efficiency and found a U-shape relationship between environmental regulation and total factor energy efficiency. The SFA model can also measure energy efficiency. Liu et al. [6] used it to measure the total factor energy efficiency of 30 provinces in China from 2000 to 2016, finding that regional energy efficiency in China presents a spatial feature of “high in the east and low in the west”. It is worth noting that Zhang et al. [7] calculated the energy resilience and efficiency values of 30 provinces from 2000 to 2019 from the perspective of coordinated development of resilience and efficiency. There are differences in the level of energy development among regions. Similarly, Gu et al. [8] used the BMA method to identify the key influencing factors of energy efficiency improvement at the national level and the eastern, central and western regions. The results show that during the 12 years, the national and regional energy efficiency shows a fluctuating upward trend whereby, on average, that of the western region has increased and those for the eastern and central regions have slightly decreased.

Second, for its influencing factors, there is a large strand of the literature exploring the impact factors of energy efficiency from the aspects of energy structure, industrial structure, government expenditure, opening up, etc., and energy efficiency closely correlates to the economic development level [9,10,11]. Li et al. [12] believed that richer city resources means greater energy efficiency. Wei [13] found via empirical analysis that marketization level stimulates energy efficiency. Liu et al. [14] used the Tobit model to study the total factor energy efficiency structure and its key influencing factors in rural areas of western China. Their results showed that changes in industrial structure, energy conservation, emission reduction, and other innovative technologies jointly enhance regional total factor energy efficiency, and that changes in traditional energy prices and a drop in the proportion of coal fuel consumption both promote increases in total factor energy efficiency. As for government intervention, Lin et al. [15] stated that China’s energy market is subject to considerable government intervention, which does not hinder the self-distribution mechanism of China’s energy market but does inhibit the improvement of energy efficiency. In this study, we focus on the impact of industrial structure on energy efficiency. With the continuous optimization of industrial structure, regional governments are developing the optimal allocation of resources according to local conditions, leading to a great difference in the level of industry between regions and to corresponding changes in the mode, type, and demand of energy consumption. On the other hand, as a “resource converter” between economic input and output, the combination and adjustment mode of industrial structure play a decisive role in energy utilization efficiency [16]. Given the impact of industrial structure on energy efficiency, the academic community has long noted that adjusting the industrial structure is an important way to improve energy efficiency and reduce the waste of resources. Some scholars found that the optimization mode of industry from heavy industry to light industry is more conducive to reducing energy consumption intensity. Wei et al. [17] drew a similar conclusion in that a rise in the proportion of tertiary industry in GDP has a positive and rising impact on improving energy efficiency. Industrial structure optimization plays a positive role in promoting energy efficiency [18]. Liu et al. [19] analyzed data from 2007 to 2016 in different time periods and regions based on the spatial Dubin model to study the influence mechanism of population size and industrial structure on energy efficiency. Their results showed that the optimization of industrial structure has a significantly positive promoting effect on energy efficiency. Some scholars have opposite conclusions. Cheng et al. [20] said that industrial structure inhibits the improvement of energy efficiency. Deng [21] used a spatial econometric model to study the energy efficiency and influencing factors of China’s prefecture-level cities. The results showed that industrial structure and energy efficiency have a strong negative effect. Guo et al. [22] analyzed the spatial evolution characteristics of China’s urban energy efficiency from 2005 to 2015 based on the two-stage Super-SBM, concluding that industrial structure and energy efficiency have changed from a significantly negative correlation to a non-significant correlation in the whole country and the central region. Zhang et al. [23] also drew similar conclusions. They used a quantile regression model to explore the influencing factors of urban energy efficiency from the perspective of efficiency differences, and the results showed that industrial structure, urbanization level and scientific and technological expenditure hindered the improvement of urban energy efficiency.

Throughout the literature of energy efficiency, a spatial relationship is found exist between regions’ energy efficiency. Most scholars used a spatial econometric model to estimate and analyze the spatiality of regional energy efficiency and its influencing factors. The above studies have laid a solid theoretical foundation for the research framework of this current paper. Therefore, by considering the spatial differences between energy efficiency, this research also utilizes a spatial model to study the mechanism of the influence of industrial structure on provincial energy efficiency. It starts from a broader perspective and fully considers the influences of government intervention, technological innovation, marketization level and average resident wage level on provincial energy efficiency, examines the spatial differences of energy efficiency, and analyzes the impact of industrial structure on energy efficiency. To sum up, this research will use the panel data of 16 years from 2004 to 2019 and the spatial econometric model method to carry on research, the main contribution of this study being as follows: (1) Since the geographical environment of different regions in China is very different, the level of industrial development is not the same. However, there may be similar industrial structures between adjacent regions, and it seems more conducive to the study of the spatial relativity of energy efficiency to divide the 30 provinces into different categories according to certain classification standards. This research will use the Moran index and Moran scatter plot to study the spatial relativity of energy efficiency in each region. Meanwhile, according to the geographical location, 30 provinces in China will be divided into eastern, central and western, and then the spatial difference of energy efficiency in these three regions is analyzed. (2) Due to the obvious spatial autocorrelation of the research objects, the spatial econometric model is more convincing than the econometric model without spatial effect, which can fully and effectively explain the influence of explanatory variables on the explained variables and the spatial spillover effect between regions. At the same time, in order to make the research conclusions more reliable, this study studies the spatial influence mechanism of industrial structure on energy efficiency from three aspects: the whole sample, different time periods and different fields (3). In addition, this research selects government intervention, technological innovation, marketization level and average resident wage level as its control variables. We will study the mechanism of their impact on energy efficiency.

## 2. Data Description and Model Setting

### 2.1. Research Regions

The research scope of this paper covers 30 provinces and cities in China (except Tibet, Taiwan, Hong Kong and Macao), among which the research field can be divided into three parts according to geographical location: eastern, central and western, as shown in Figure 1.

### 2.2. Data Sources and Description

Explained variable: This paper takes the GDP per unit of energy consumption to measure the comprehensive energy efficiency value of different regions. GDP per unit energy consumption is calculated as follows:
*En* = GDP/Energy Consumption (primary energy converted into standard coal)


Here, *En* is energy efficiency. A higher value implies a greater energy efficiency for a region. A smaller value means a smaller energy efficiency for a region.

Explanatory variable: The upgrading of industrial structure is a dynamic indicator of continuous change, but also an important indicator to measure the economic strength of a country or region. Industrial structure shows two essential characteristics: first, the changing relationship between industrial proportions; second, the productivity of the industrial sector has improved [24]. Considering the rapid development of China’s secondary industry and tertiary industry, this study adopts the ratio of service industry to industrial industry to reflect the development trend of the industrial sector. Figure 2 shows the trend of industrial structure changes over time in the three eastern, central and western regions.

Control variables: To reduce any missing variables from affecting energy efficiency as much as possible, the following four control variables are selected herein according to the literature. (1) Government intervention (*Gove*) is a national macro-control means for the overall social economy. This study measures it by the ratio of the annual fiscal expenditure of the local government to the local GDP of that year. (2) Technological innovation (*Seva*) is an important source for enterprises and countries to enhance their competitiveness, and also an important factor affecting modern economic activities. Industrial technological innovation can shorten the production process and reduce the intermediate cost consumption, which can further improve energy efficiency to a certain extent [25]. This study uses the ratio of local financial expenditure on science and technology to the total amount of general local financial expenditure to measure technological innovation in various regions. (3) Marketization level (*Mark*) builds a social resource distribution platform through the market mechanism, helping to promote the optimal allocation of resources, increase the circulation speed of factors and thus affect energy utilization efficiency. The study uses the ratio between the total capital of private industrial enterprises and the total assets of industrial enterprises at a certain scale and above to measure Mark. (4) Average resident wage level (*Wage*) is an important national livelihood issue. This study uses the average salary level of urban employed personnel to measure *Wage*.

There are 31 provinces and cities in China, but since the energy consumption and energy production of Xizang are relatively low, or it even lacks of energy data for most years, Xizang is not included in the analysis. The study thus selects, panel data of the remaining 30 provinces and cities from 2004 to 2019 as the research object for spatial econometric model analysis. All data are from the National Bureau of Statistics of China, China Statistical Yearbook (2005–2020), and China Energy Statistical Yearbook (2005–2020).

### 2.3. Model Setting and Spatial Econometric Model

This research sets the model as:$$En=\;f\left(Insr,\;Gove,\;Seva,\;Mark,\;Wage,\;\varepsilon \right)$$

Here, *En* represents energy efficiency, *Insr* represents industrial structure, *Gove* represents government intervention, *Wave* represents average resident wage level, *Seva* represents technological innovation, *Mark* represents marketization level, and *ε* represents the residual item.

The spatial autocorrelation test can reveal the spatial relationship of an individual characteristic. Before applying the spatial autocorrelation test, there is no clear and reliable prior information to tell us about the spatial dependence direction for the existence of space units. When the arrangement of space units is irregular, judging their spatial relationships becomes particularly complex, and so selecting an appropriate and effective spatial weight matrix is the first step of and foundation for spatial exploration and analysis verification. Therefore, from the general applicability of the spatial weight matrix theory, this study adopts the inverse space distance weight matrix. The spatial distance weight matrix can overcome the dependence on spatial direction and reduce research errors caused by complex spatial rules more effectively. The expression of space distance matrix is as follows:(1)$$W_{ij}=\left\{\begin{array}{cc} \frac{1}{d_{ij}} & i\neq j\left(d_{\mathrm{ij}}\;\mathrm{represents}\;\mathrm{the}\;\mathrm{straight}-\mathrm{line}\;\mathrm{distance}\;\mathrm{between}\;\mathrm{region}\;i\;\mathrm{and}\;\mathrm{region}\;j\right)\\ 0 & i=j \end{array}\right.$$

This research uses the average aggregation degree of global Moran’s I response spatial variables to test whether energy efficiency is similar, different or independent in the whole region [26]. The expression is:(2)$$\mathrm{Moran}\prime \mathrm{sI}=\frac{\sum_{i=1}^{n}\sum_{i\neq j}^{n}w_{ij}\left[x_{i}-\tilde{x}\right]}{S^{2}\sum_{i=1}^{n}\sum_{j=1}^{n}w_{ij}}$$

Moran’ s I value ranges between −1 and 1. If Moran’ I is greater than zero, then the spatial similarity of the two regions positively correlates; If Moran’ I is less than zero, then the spatial similarity of the two regions negatively correlates. The closer the index value is to 1 or −1, the higher the region’s spatial similarity is. If the index equals 0, there is almost no spatial correlation between regions.

Local Moran’s I is used to estimate regional heterogeneity, and the commonly used analysis tool is the Moran scatterplot. The expression of local Moran’s I is:(3)$$I_{I}=\frac{\left(x_{i}-\;x\;¯\right)\sum_{j=1}^{n}W_{\mathrm{ij}}\left(x_{i}-\;x\;¯\right)}{S^{2}}$$

Scatter points in the Moran scatterplot are put into four quadrants according to their different statistical properties. A spatial positive autocorrelation is expressed as HH (High–High, high values are adjacent to high values) or LL (Low–Low, the low value is adjacent to low value), that is, the scatter points are located in the first and third quadrants. A spatial negative correlation is expressed as HL (High–Low, high values are adjacent to low values) or LH (Low–High, the low value is adjacent to high value), that is, the scatter points are located in the second and fourth quadrants.

The spatial econometric model takes into account the spatial correlation and spatial heterogeneity among panel data, and so its analysis and design ideas and numerical prediction research conclusions are more scientific and reasonable [27]. The classical spatial econometric analysis models include: spatial lag model (SAR), spatial Durbin model (SDM) and spatial error model (SEM). SDM can be degenerated into SAR and SEM, and its simple arithmetic expression is as follows:y = ρWy + Xβ+ WXγ + ε(4)

In the expression, y represents the explanatory variables, X represents the explanatory variables, W is the space weight matrix, ε represents the error term, and ρ and γ are both space lag parameters. The above spatial lag parameters can be used to select an appropriate spatial measurement model: when γ = 0 and ρ ≠ 0, SDM can directly degenerate into SAR; when γ =-ρβ, SDM can completely degenerate into SEM.

## 3. Empirical Analysis

### 3.1. Spatial Correlation Test

Before quantitative analysis, global Moran’ s I is first used to observe the spatial distribution of energy efficiency, and present the estimated results in Table 1. According to the comparison and analysis results of the global spatial autocorrelation test, conducted from 2004 to 2019, the Moran’ I of all provinces in China is positive, and the values all pass the significance test of less than 1%, respectively (*p* < 0.01), indicating positive spatial dependence of a significant degree between energy efficiency. To test local spatiality, the Moran scatter plot is used to investigate the local correlation of provincial energy efficiency in China.

Figure 3 shows the Moran scatterplot of the energy efficiency of China’s provinces in 2004, 2009, 2014, and 2019, the majority of them are at a high energy efficiency level when surrounded by other highly energy efficient provinces. Shanghai, Jiangsu, Zhejiang, Anhui, Fujian, Jiangxi, Hunan, Guangdong, and Hainan are stable in the first quadrant (High–High) all the time. These provinces are located in the southeast coastal region and surrounding region, forming a concentrated area of high energy efficiency in China. After 2004, Hubei joined the ranks. Although there is higher energy efficiency in these provinces, the disadvantage is that they also have a high spatial lag. Heilongjiang, Jilin, Hebei, Liaoning, Shanxi, Inner Mongolia, Yunnan, Gansu, Qinghai, Ningxia, and Xinjiang are in the third quadrant (Low–Low). These provinces can be divided into two categories according to their unique characteristics. One is the northwest region represented by Yunnan and the southwest region represented by Gansu and Xinjiang, whose economic base, technical strength and degree of opening to the outside world are lower than the national average. The other type of provinces are those with energy consumption and real economy as their pillar, such as Shanxi and Inner Mongolia. These provinces mainly use coal as energy, showing characteristics of extensive economic development, and their distribution of industrial structure is not balanced, thus restricting economic development. Shandong and Guizhou remain in the second quadrant (Low–High). They are mainly characterized by low energy efficiency and high spatial lag. These provinces have low energy efficiency but are adjacent to highly energy efficient provinces. Beijing always stays within the fourth quadrant (High–Low). After 2004, Chongqing entered the fourth quadrant. The main performance here is high energy efficiency and low spatial lag. Although their energy efficiency is the highest, they have a relatively large gap with the energy efficiency level of surrounding provinces and cities, which is unique in surrounding provinces. This phenomenon shows that their own high energy efficiency has not spread to the surrounding areas.

Based on the above findings, it is concluded that there are regional spatial differences in energy efficiency among provinces in China and there are large local spatial distribution differences, it is consistent with the conclusion of Wang [28]. Therefore, according to the regional differences in energy efficiency levels, the mechanism of industrial structure on energy efficiency is studied from the correlation and heterogeneity of spatial dimensions.

### 3.2. Spatial Econometric Analysis

#### 3.2.1. A Study of Provincial Energy Efficiency in China under Full Sample Conditions

Under the condition of the full sample, whether the spatial metrology model fits the data needs to pass the LM test. In the four cases of LM test, LM-ERR, R Lm-ERR and LM-LAG all pass the significance test of 1%, indicating spatial correlation between the energy efficiency of China’s provinces. Therefore, the study herein uses spatial models more effectively than those without spatial ones. If it is true when Hausman tests the null hypothesis $H_{0}=0$, then it means that the random effects model is selected. The statistics of the test results H can judge that the null hypothesis of the test is rejected when the significance level is lower than 1%, and the fixed effects model should be adopted for the selection of the model. The results of the two LR tests are, respectively, shown in Table 2. LR(SAR) and LR(SEM) and LR (ind) and LR (time) pass the 1% significance test, indicating that the two reject the original hypotheses of SAR and SEM and the original hypothesis of using the individual fixed effects model and the time fixed effects model. Therefore, through the analysis of the model, the dual fixed effect spatial Durbin Model will be used.

Table 3 reports the regression results of the spatial Durbin model under full sample conditions. According to the results of the regression, rho and sigma2_e values are positive and pass the significance test. This means that the levels of industrial structure and energy efficiency in China in the provincial distribution index still exhibit a positive spatial correlation, and the energy efficiency of industrial structure to the adjacent area has significant spatial spillover effects. In the regression results of SDM, the *Insr* coefficient value is 0.297, passing the 1% significance test. The industrial structure and energy efficiency have a positive relationship, indicating that the greater the proportions of the total output value of the tertiary industry and the total output value of the secondary industry are, the more beneficial they will be to the improvement of energy efficiency. The coefficient value of *W* × *Insr* is −1.386, indicating that the industrial structure has a negative spillover effect on the surrounding areas. Therefore, the area with a large secondary industry has a promoting effect on the energy efficiency of the surrounding provinces.

The model also introduces government intervention, technological innovation, marketization level and average resident wage level as control variables. According to the regression results of SDM, the *Gove* coefficient is 0.433 and passes the 5% significance level test. This shows that government intervention and energy efficiency have a negative relationship, as the government exhibits excessive intervention behavior to promote energy efficiency. Because solving the energy efficiency problem is a long-term process, driven by the inertia of China’s extensive consumption, government officials pay more attention to achieving short-term economic goals in order to try and achieve economic growth goals more quickly. Such behavior by officials will “infect” surrounding areas, especially areas with similar market conditions. They are reluctant to become “pioneers” and more willing to become followers. The coefficient of *Seva* is significantly positive. This indicates a positive correlation between scientific and technological innovation and energy efficiency. Moreover, the level of science and technology has a long-term spreading effect. The promotion of scientific production technology can effectively improve energy consumption technology so as to improve provincial energy efficiency. *Mark* and *Wage* do not pass the significance level test. This indicates that the marketization level and average household income have little influence on energy efficiency. The specific direction of influence cannot be determined.

#### 3.2.2. China Provincial Energy Efficiency Study by Different Time Period

Considering the time-lag effect of China’s macroeconomic management, the previous policy plan may have an impact on the policy implementation in the next stage, and the energy efficiency of each province has a trend of time difference. As such, this paper divides the samples into two periods: 2004–2011 and 2012–2019. To study the impact of industrial structure on energy efficiency in different periods. The spatial effect of the two periods is tested by the spatial autocorrelation test and the LM test and LR test of whether the spatial Durbin model can be reduced to the spatial lag model and the spatial error model. The estimation results of Table 4 also show that the spatial Durbin model with double fixed effects should be selected for both periods.

Table 5 lists the estimation results of the spatial model by different time periods. According to the estimation results of the spatial Durbin model with double fixed effects, the expansion of the industrial structure from 2004 to 2011 promoted the improvement of energy efficiency, while the impact of industrial structure on energy efficiency is not significant during 2012–2019. The government expects to improve energy efficiency by optimizing the industrial structure. The possible reason for this result is that in the early stage, the government expected to improve energy efficiency by optimizing the industrial structure, while in the later stage, due to the increasingly complex social economy, the promotion mechanism of energy efficiency became diversified, and industrial structure was not the only way to improve energy efficiency. The effects of government intervention in the two periods are opposite. The impact of Mark on energy efficiency in the second stage is significantly positive, indicating that marketization assisted by good policies can bring about the optimal utilization of resources. Scientific and technological innovation plays a significantly positive role in the previous stage. *Wage* has a significantly negative impact on the improvement of residents’ income level in the later stage, increasing the social consumption structure and promoting the diversification of industries.

#### 3.2.3. China Provincial Energy Efficiency Study by Different Regions

The mechanism of energy efficiency is different in each province and so it is necessary to estimate the spatial econometric regression separately for the eastern, central and western regions. Through the model test of different regions and combined with the experience of model selection described above, according to the results in Table 6 the spatial Durbin model with double fixed effects should be used for the empirical analysis of the eastern, central and western areas.

Table 7 reports the regression results of different regions under the three different effects. The research focuses on the estimation results of the spatial econometric model suitable for the three regions, and mainly studies the mechanism and difference of the impact of industrial structure on the energy efficiency of the three regions. According to the regression results, the industrial structure coefficient in the eastern region is positive and passes the significance test of 1%, and so the industrial structure in the eastern region promotes the improvement of energy efficiency. This conclusion is also consistent with the spatio-temporal distribution of energy efficiency in China’s provinces. The southeast coastal area has a concentration of talents, a relatively developed economic structure, a rapid development of the service industry, a more reasonable industrial structure, and a relatively high energy utilization efficiency. Especially affected by the adjustment of the national strategy in 2012, the energy efficiency level of the east region is significantly higher than the national average level. This is determined not only by the quality of regional economy, but also by the ecological environment itself. For example, Hainan is an international tourism island supported by the central government. With tourism, service industry and other tertiary industries acting as the pillars of its economy, Hainan has made great contributions to local economic development. In addition, Hainan also performs better in energy efficiency according to the analysis in the Moran scatterplot above. This is because Hainan is a region with tertiary industry and service industry as the main parts of its economy. The economic system is characterized by less energy consumption and higher economic added value, and so the resulting energy efficiency is also higher. 

The industrial structure in central and western China has a significantly negative impact on energy efficiency. In other words, the smaller the ratio between the tertiary industry and the secondary industry, the more energy efficiency can be promoted. In recent years, secondary industry has developed relatively well in the central region, but the development of tertiary industry is relatively scarce. According to the estimation results of the full sample spatial metrology model, optimization of the industrial structure mainly starts with the development of the tertiary industry, which can effectively promote the improvement of energy efficiency. However, there is a lack of more effective measures for the unbalanced resource distribution of the secondary industry, and it is difficult to offset the negative effect of the secondary industry on energy efficiency by promoting the development of the tertiary industry. As part of the central region, Shanxi’s disadvantage is more obvious. Heavy industry is a pillar industry in most of the central region. Coal is the main energy consumption, and there are more polluting and large industrial enterprises. The three industrial structure is unbalanced, resulting in relatively low efficiency of energy use. It can be seen from the empirical results that the optimization of industrial structure can promote improvement of energy efficiency, but due to large differences in the level of development between provinces, the impact on energy efficiency is also different. The west region is mainly dominated by agriculture, and the secondary and tertiary industries are relatively backward compared with the east and central regions. Therefore, the west region needs to reduce energy consumption and improve efficiency under the rational economic distribution of the secondary and tertiary industries.

## 4. Conclusions and Implications

Based on the spatial econometric model, the results herein are as follows. 

First, there are significant spatial correlation differences and local spatial differences in China’s provincial energy efficiency.

Second, in the full sample condition, the optimization and upgrading of the industrial structure has a positive effect on increasing energy efficiency. That is, the greater the proportions of the total amounts of the tertiary industry and the secondary industry are, the higher the energy efficiency level will be; but the negative spatial spillover effect of the industrial structure inhibits the energy efficiency level of surrounding regions. Government intervention has negative effects on energy efficiency and technological innovation has beneficial effects on energy efficiency. Marketization level and average resident income have negligible effect on energy efficiency. The government needs to more actively promote the economic transformation of the industrial structure so as to achieve environmentally friendly and energy economy development goals. China should focus on independent innovation, guide the flow of talents, capital, technology, and other production factors to the tertiary industry, eliminate backward production capacity with low efficiency and high consumption, continuously improve the conversion rate of scientific and technological achievements, actively cultivate a modern service industry, improve the level and management ability of the service industry, and promote the intensive use of clean resources [29]. There are some problems in China’s domestic market, such as the structure being unreasonable, the prices not being perfect. At the same time, the government intervenes too much in market behaviors, restricting freedom in market resources and the rational distribution of energy efficiency improvement. The central government should minimize direct control of market cost pricing, the market energy production plan, and market power pricing, reduce indirect control of links within energy market resource allocation, deepen the reform of the national energy market economic system, and gradually integrate with the current international energy market.

Third, the empirical analysis is carried out by time and region, and the results show that the industrial structure has a positive effect on energy efficiency during 2004–2011. The industrial structure in the east promotes the improvement of energy efficiency, while the central and western regions have a negative effect on energy efficiency. From the characteristics of energy structure, government investment, and other aspects, energy policies should be formulated according to local conditions, differences between provinces should be narrowed, and the characteristic of coal energy as the main energy consumption in the central region should be changed for the better. The government and industrial enterprises should promote the burning of clean coal, the diversification of energy consumption, the reform of energy production and consumption system, and the improvement of energy efficiency by strengthening technological innovation or introducing foreign advanced technology. There are still some regions, especially resource-based cities, whose own characteristics of energy endowment are deeply rooted in the local industrial production stage. Therefore, it is not feasible to just copy the development experience of the east region; they need to fully understand their own resource endowment and formulate appropriate and reliable adjustment policies based on their own local development situation.

## Figures and Tables

**Figure 1 ijerph-20-02134-f001:**
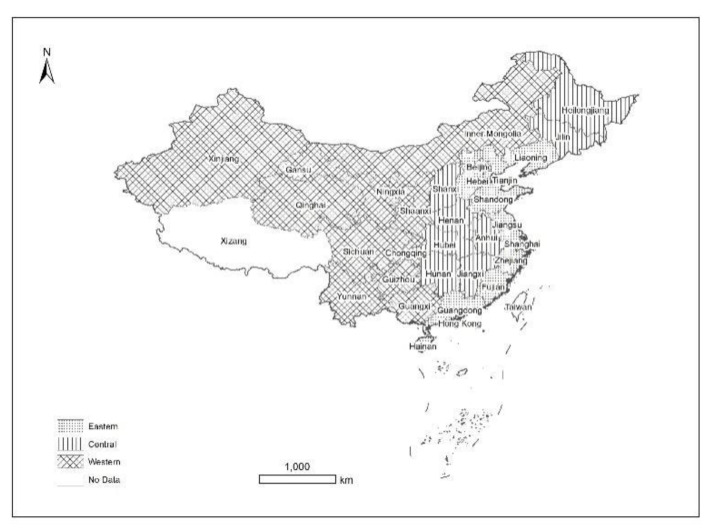
Schematic diagram of the three regions of China.

**Figure 2 ijerph-20-02134-f002:**
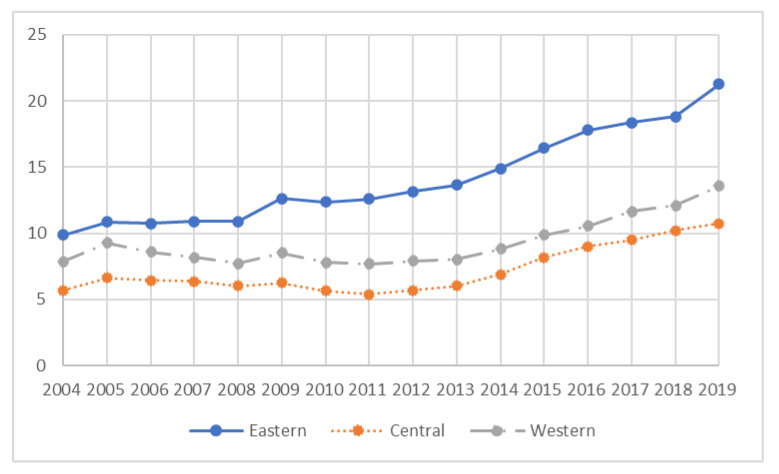
Change trend of industrial structure in three regions of China.

**Figure 3 ijerph-20-02134-f003:**
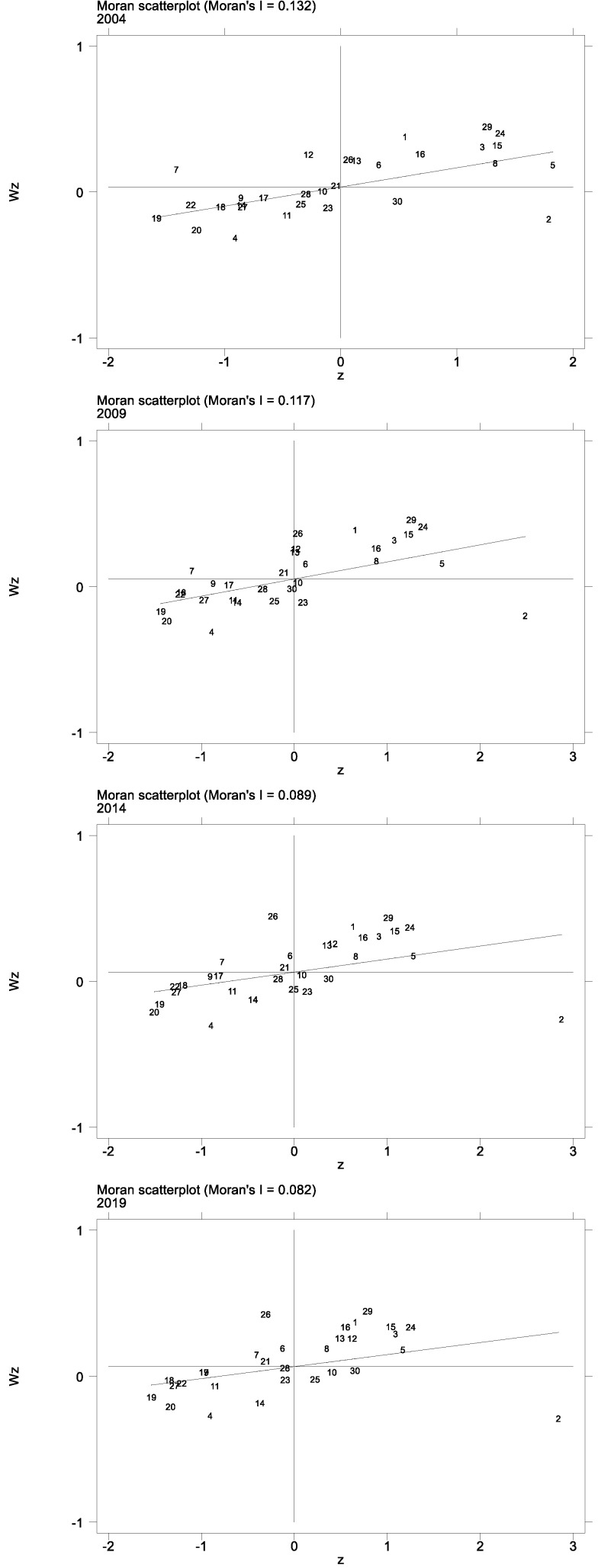
Moran scatter plot of energy efficiency for 2004, 2009, 2014 and 2019. Note: Anhui, Beijing, Fujian, Gansu, Guangdong, Guangxi, Guizhou, Hainan, Hebei, Henan, Heilongjiang, Hubei, Hunan, Jilin, Jiangsu, Jiangxi, Liaoning, Inner Mongolia, Ningxia, Qinghai, Shandong, Shanxi, Shaanxi, Shanghai, Sichuan, Tianjin, Xinjiang, Yunnan, Zhejiang, and Chongqing are replaced by the numbers 1–30, respectively.

**Table 1 ijerph-20-02134-t001:** The global Moran’s I index of inter-provincial energy efficiency and its test.

Year	Moran’ I	Z	*p*-Value
2004	0.132	4.720	0.000
2005	0.131	4.731	0.000
2006	0.125	4.563	0.000
2007	0.117	4.357	0.000
2008	0.115	4.302	0.000
2009	0.117	4.359	0.000
2010	0.114	4.269	0.000
2011	0.109	4.133	0.000
2012	0.104	3.998	0.000
2013	0.088	3.578	0.000
2014	0.089	3.618	0.000
2015	0.081	3.358	0.000
2016	0.079	3.316	0.000
2017	0.084	3.438	0.000
2018	0.088	3.560	0.000
2019	0.082	3.407	0.000

**Table 2 ijerph-20-02134-t002:** Model selection test.

	Test Value
LM-err	129.643 ***
R LM-err	82.264 ***
LM-lag	48.452 ***
R LM-lag	1.072
Hausmann test	22.63 **
LR (SAR)	66.20 ***
LR (SEM)	60.96 ***
LR (ind)	59.70 ***
LR (time)	417.81 ***

Note: “***” and “**” respectively indicate passing the significance level test of 1% and 5%.

**Table 3 ijerph-20-02134-t003:** Spatial model estimation results.

Variable	SDM		
	sFE	tFE	stFE
*Insr*	0.356 *** (6.14)	0.454 *** (8.72)	0.297 *** (4.62)
*Gove*	−0.564 *** (−2.94)	−1.675 *** (−9.83)	−0.433 ** (−2.15)
*Seva*	6.531 *** (3.60)	9.881 *** (5.09)	5.619 *** (3.16)
*Mark*	0.236 (0.62)	0.781 *** (2.66)	0.483 (1.24)
*Wage*	0.349 (1.62)	0.226 ** (1.98)	0.117 (0.53)
*W* × *Insr*	−0.551 *** (−2.90)	−0.552 * (−1.81)	−1.386 *** (−3.86)
*W* × *Gove*	0.352 (1.33)	2.644 ** (−9.83)	2.059 ** (2.04)
*W* × *Seva*	−11.487 *** (−2.77)	9.881 *** (5.09)	24.081 ** (2.51)
*W* × *Mark*	0.108 (0.07)	8.869 *** (4.23)	12.001 *** (3.54)
*W* × *Wage*	−0.021 (−0.07)	−2.465 *** (−3.74)	−0.552 (−0.37)
$R^{2}$	0.6664	0.1207	0.1347
Rho	0.613 *** (8.48)	0.541 *** (5.48)	0.370 *** (3.03)
sigma2_e	0.049 *** (14.27)	0.105 *** (15.23)	0.044 *** (15.36)
Log-likelihood	35.7509	−143.3035	65.6019

Notes: “***”, “**”, and “*”, respectively indicate passing the significance level test of 1%, 5% and 10%. The values in brackets are Z-values. sFE stands for individual fixed effect, tFE stands for time fixed effect, and stFE stands for spatiotemporal dual fixed effect.

**Table 4 ijerph-20-02134-t004:** Model selection test.

	Test Value	
	**2004–2011**	**2012–2019**
LM-err	85.442 ***	26.052 ***
R LM-err	51.862 ***	28.049 ***
LM-lag	33.728 ***	5.142 **
R LM-lag	0.149	6.938 ***
Hausmann test	25.23 **	17.93 *
LR (SAR)	47.58 ***	54.05 ***
LR (SEM)	47.57 ***	54.63 ***
LR (ind)	45.01 ***	22.74 ***
LR (time)	615.21 ***	544.69 ***

Note: “***”, “**: and “*”, respectively indicate passing the significance level test of 1%, 5%, and 10%.

**Table 5 ijerph-20-02134-t005:** Regression results of spatial models in different periods.

	2004–2011			2012–2019		
variable	stFE	sFE	tFE	stFE	sFE	tFE
*Insr*	0.098 *** (−0.04)	0.183 *** (−0.03)	0.473 *** (−0.05)	0.123 (−0.08)	0.165 ** (−0.07)	0.438 *** (−0.10)
*Gove*	−0.276 *** (−0.07)	−0.365 *** (−0.07)	−1.119 *** (−0.15)	0.738 *** (−0.21)	0.853 *** (−0.20)	−2.107 *** (−0.28)
*Mark*	−0.06 (−0.24)	0.332 (−0.24)	1.160 *** (−0.27)	0.582 (−0.43)	0.456 (−0.42)	0.765 (−0.54)
*Seva*	4.2408 *** (−0.69)	4.7701 *** (−0.73)	5.4225 *** (−2.00)	−6.700 *** (−2.30)	−7.421 *** (−2.28)	12.270 *** (−2.94)
*Wage*	−0.0573 (−0.09)	−0.0133 (−0.09)	0.1267 (−0.09)	1.8909 *** (−0.34)	1.5946 *** (−0.34)	0.4726 ** (−0.23)
*W* × *Insr*	−1.0351 *** (−0.20)	−0.393 *** (−0.10)	0.6320 ** (−0.32)	−1.6202 *** (−0.52)	−1.5531 *** (−0.26)	−0.705 (−0.60)
*W* × *Gove*	−0.3642 (−0.36)	0.7092 *** (−0.13)	2.8808 *** (−0.90)	6.3839 *** (−1.15)	2.4737 *** (−0.67)	1.8923 (−1.75)
*W* × *Mark*	1.0895 (−1.95)	2.6546 ** (−1.16)	10.7522 *** (−1.72)	1.5025 (−3.88)	0.4698 (−1.65)	7.8990 * (−4.41)
*W* × *Seva*	18.3029 *** (−4.02)	−2.6186 * (−1.52)	3.1856 (−11.58)	−28.327 * (−15.12)	−27.151 *** (−10.01)	19.8748 (−19.79)
*W* × *Wage*	0.3538 (−0.68)	−0.0767 (−0.15)	−1.0398 ** (−0.52)	4.2888 * (−2.50)	2.4790 *** (−0.71)	−4.7874 *** (−1.60)
60rho	−0.3044 (−0.24)	0.4185 *** (−0.12)	0.4178 ** (−0.18)	−0.4756 * (−0.27)	−0.1068 (−0.23)	0.5477 *** (−0.14)
sigma2_e	0.0027 *** (−0.0002)	0.0032 *** (−0.0003)	0.0346 *** (−0.0032)	0.0156 *** (−0.0014)	0.0175 *** (−0.0016)	0.1516 *** (−0.0141)

Notes: “***”, “**”, and “*”, respectively indicate passing the significance level test of 1%, 5% and 10%. The values in brackets are Z-values. sFE stands for individual fixed effect, tFE stands for time fixed effect, and stFE stands for spatiotemporal dual fixed effect.

**Table 6 ijerph-20-02134-t006:** Model selection in different regions.

	Eastern	Central	Western
LM-err	15.236 ***	119.569 ***	163.524 ***
R LM-err	15.726 ***	51.402 ***	12.148 ***
LM-lag	2.471 *	73.735 ***	187.237 ***
R LM-lag	2.961 *	5.568 **	35.861 ***
Hausmann test	73.09 ***	20.31 ***	65.7 ***
LR (SAR)	32.09 ***	38.72 ***	58.17 ***
LR (SEM)	43.93 ***	11.64 **	63.40 ***
LR (ind)	55.28 ***	94.91 ***	46.08 ***
LR (time)	184.28 ***	209.47 ***	224.11 ***

Note: “***”, “**” and “*”, respectively indicate passing the significance level test of 1%, 5%, and 10%.

**Table 7 ijerph-20-02134-t007:** Regression results of spatial models in different regions.

Variable	Eastern			Central			Western		
	stFe	sFe	tFe	stFe	sFe	tFe	stFe	sFe	tFe
*Insr*	0.327 *** (−0.08)	0.566 *** (−0.09)	0.093 (−0.11)	−0.52 *** (−0.09)	−0.386 *** (−0.09)	0.094 (−0.12)	−0.490 *** (−0.17)	−0.330 *** (−0.13)	−0.500 ** (−0.25)
*Gove*	−0.530 (−0.38)	−0.194 (−0.43)	−2.556 *** (−0.58)	1.119 * (−0.62)	0.282 (−0.74)	2.264 *** (−0.77)	−0.242 (−0.21)	−0.343 * (−0.20)	−2.509 *** (−0.28)
*Mark*	−0.455 (−0.57)	−0.230 (−0.65)	−1.474 ** (−0.66)	1.694 *** (−0.46)	1.544 *** (−0.54)	6.080 *** (−0.67)	−2.084 ** (−0.84)	−3.028 *** (−0.63)	−2.275 ** (−1.09)
*Seva*	4.325 * (−2.39)	−0.786 (−2.52)	19.207 *** (−3.06)	−2.398 (−1.74)	−1.954 (−2.53)	13.865 *** (−3.18)	0.094 (−4.22)	0.559 (−3.87)	24.280 *** (−7.08)
*Wage*	−0.472 (−0.32)	−0.121 (−0.32)	0.234 (−0.20)	1.436 *** (−0.23)	1.306 *** (−0.32)	0.742 (−0.51)	0.635 (−0.41)	0.446 (−0.33)	−1.610 ** (−0.653)
*W* × *Insr*	−1.161 *** (−0.24)	−0.820 *** (−0.21)	−1.393 *** (−0.30)	−0.416 * (−0.22)	0.373 ** (−0.15)	1.253 *** (−0.31)	−2.845 *** (−0.80)	−0.931 *** (−0.29)	−2.509 * (−1.36)
*W* × *Gove*	−2.292 (−1.81)	−0.850 (−0.52)	1.380 (−2.85)	2.126 (−1.98)	−0.808 (−0.76)	10.225 *** (−2.34)	−0.982 (−1.05)	−0.412 (−0.25)	−6.284 *** (−1.39)
*W* × *Mark*	−4.447 (−2.73)	−5.331 *** (−1.69)	−5.210 *** (−1.92)	5.267 *** (−1.60)	−0.747 (−1.13)	14.065 *** (−2.42)	−5.518 (−4.43)	−10.274 *** (−2.63)	−10.456 * (−5.57)
*W* × *Seva*	20.023 ** (−9.40)	−7.303 ** (−3.71)	64.393 *** (−13.38)	−0.297 (−5.63)	1.672 (−5.60)	−45.677 *** (−11.54)	−33.021 (−30.79)	−26.950 *** (−8.86)	53.180 (−53.89)
*W* × *Wage*	0.671 (−1.15)	2.129 *** (−0.38)	−1.742 * (−0.94)	3.614 *** (−1.08)	−0.909 *** (−0.34)	7.426 *** (−1.81)	7.949 *** (−2.21)	1.087 ** (−0.46)	−3.102 (−2.94)
Rho	−0.996 *** (−0.17)	−0.338 ** (−0.15)	−0.089 (−0.16)	−0.764 *** (−0.16)	0.467 *** (−0.09)	−0.379 ** (−0.19)	0.187 (−0.17)	0.408 *** (−0.13)	0.222 (−0.17)
sigma2_e	0.028 *** (−0.003)	0.044 *** (−0.005)	0.092 *** (−0.010)	0.009 *** (−0.001)	0.020 *** (−0.003)	0.051 *** (−0.006)	0.023 *** (−0.003)	0.029 *** (−0.003)	0.081 *** (−0.009)
N	176	176	176	128	128	128	176	176	176
R^2^	0.332	0.266	0.175	0.541	0.674	0.534	0.327	0.184	0.249

Notes: “***”, “**”, and “*”, respectively indicate passing the significance level test of 1%, 5% and 10%. The values in brackets are Z-values. sFE stands for individual fixed effect, tFE stands for time fixed effect, and stFE stands for spatiotemporal dual fixed effect.

## Data Availability

Not applicable.

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
