# Peer review of "Analysis on the Influence of Industrial Structure on Energy Efficiency in China: Based on the Spatial Econometric Model"

_ijerph, 2023, doi:10.3390/ijerph20032134_

Round 1

Reviewer 1 Report

1.     In Introduction, contribution is not clear. What kind of research gap is fulfilled by the proposed method? Is any shortcoming of the existing methods overcome by the proposed method?

2.     The format of variables must be unified in the whole paper. Namely, En, Insr etc. should be italic.

3.     In Table I, z and p are not defined. Besides, discussion is lacking. What do I and z reduce as time goes by mean?

4.     Since provinces in China are represented by numbers in Figure 1, It should be clarified before.

5.     The authors classify the provinces in the third quadrant by their geographic area, which is hard for the readers to understand who are not familiar with geography of China. Therefore, maps are suggested to be demonstrated.

6.     The industry structure for each province is not present using data, e.g., what is the share of secondary industry and tertiary industry in the whole industry?

Reviewer 2 Report

manuscripit written in well form need some minor correction before acceptance

1. in table "Table 1. The global Moran's I index of inter-provincial energy efficiency and its test" author did only results upto 2019 there is missing of 2020-2022 provide data upto 2022

2. there are improper wording used in manuscript such as "We use" as in reserach writing there is no use of I, We, he, or she. the author should avoid these mistakes

3. author should enhance the manuscript via adding latest literature in manuscripit.

4. carefull proof reading is needed

Reviewer 3 Report

Overall problem

The details of the articles are often wrong, and there are some problems with the format of the formulas There is less process from model building to validation Some quotations are too old, so try to use new ones  

Chapter problem

Introduction section

Too much-segmented logic is not clear enough. You can merge some paragraphs and sort out logical relations appropriately.

Literature review section

There is some logical progress, but the references are too old.

Discussion section

Insufficient innovation and contribution to research on limitations and future trends; The process of hypothesis testing is too general and the argument is not comprehensive enough.

Conclusion

The summary of the article is messy, and there should be more space for the verification of model building

Round 2

Reviewer 1 Report

The reviewer's comments have been addressed.

Reviewer 3 Report

The manuscript has improved enough on the questions I raised. I think that can be published.